# The Moderating Role of Emotional Intelligence on the Relationship Between Nurses’ Preparedness to Care for COVID-19 Patients and Their Quality of Work Life

**DOI:** 10.3390/bs14121166

**Published:** 2024-12-05

**Authors:** Hanan A. Alkorashy, Aisha F. Basheer, Hanem F. Mohamed

**Affiliations:** 1Nursing Administration and Education Department, College of Nursing, King Saud University, Riyadh 12372, Saudi Arabia; 2Nursing Administration Department, Maternity and Children Hospital, Ministry of Health, Makkah 24246, Saudi Arabia; afbasheer@moh.gov.sa; 3College of Nursing, King Saud bin Abdulaziz University for Health Sciences (KSAU-HS), Riyadh 11481, Saudi Arabia; mohamedh@ksau-hs.edu.sa; 4Medical Surgical Nursing Department, College of Nursing, Tanta University, Tanta 31527, Egypt; 5King Abdullah International Medical Research Center (KAIMRC), Ministry of National Guard Health Affairs, Riyadh 21423, Saudi Arabia

**Keywords:** emotional intelligence, nurses, COVID-19 pandemic preparedness, sickness impact profile, work–life balance, life change events, well-being

## Abstract

Emotional intelligence (EI) is increasingly recognized as a key factor in healthcare, where managing emotions is vital for job satisfaction, productivity, and interpersonal relationships. For nurses, particularly during the COVID-19 pandemic, EI plays a pivotal role in navigating emotional challenges and improving their quality of work life (QoWL). This study examined how EI moderates the relationship between nurses’ preparedness to care for COVID-19 patients and their QoWL. A cross-sectional, correlational design was used, involving 267 nurses from various healthcare settings. Data were collected through the Emotional Intelligence Scale, the Quality of Nursing Work Life survey, and demographic questionnaires. The sample was predominantly female (94.4%), with a mean age of 37.47 years (SD = 8.09) and an average of 8.43 years of experience (SD = 6.33). Most nurses (87.3%) attended COVID-19-related workshops, with 76.4% feeling prepared to care for COVID-19 patients. Emotional intelligence levels were high, with 93.6% of nurses reporting good personal competence and 85.4% reporting good social competence. Descriptive results showed that 71% of nurses perceived their QoWL as good, while 29% rated it as fair. Pearson correlation analysis revealed significant positive correlations between both personal competence (r = 0.33, *p* < 0.001) and social competence (r = 0.34, *p* < 0.001) with QoWL, but preparedness to care for COVID-19 patients did not correlate significantly with either EI or QoWL. Hierarchical regression analysis demonstrated that, although nurses’ preparedness alone did not predict QoWL (β = 0.034, *p* = 0.57), including emotional intelligence as a moderator explains 41% of the variance in QoWL. Both personal (β = 0.578, *p* < 0.001) and social competence (β = 0.665, *p* < 0.001) components of EI had significant buffering effects on the relationship between preparedness and QoWL. These findings suggest that fostering EI in nurses can enhance their resilience and improve their work life quality, particularly in high-stress healthcare environments like those experienced during the pandemic.

## 1. Introduction

Nurses’ emotional intelligence (EI) plays a critical role in shaping their interpersonal relationships and decision-making, ultimately influencing the quality of patient care. EI is a construct defined by the ability to recognize, understand, manage, and utilize emotions effectively in oneself and others. Goleman’s model of emotional intelligence comprises four primary skills: self-awareness, self-management, social awareness, and relationship management [1]. These skills provide a foundational framework that equips healthcare professionals to manage workplace stress and foster collaborative relationships effectively. In this model, both self-awareness and self-management components focus on the person, while social awareness and relationship management are concerned with the individuals’ relationships [2,3].

Although emotional intelligence does not naturally increase with age, anyone can develop it at any stage of their life [4]. Developing emotional intelligence is particularly vital in nursing, where high-stress environments and emotionally charged situations are common [5], as it helps manage workplace stress, improve interpersonal relationships, and enhance overall well-being. Research indicates that EI contributes to better decision-making [6], effective stress management [7], well-being [8], performance [9], and improved quality of work life (QoWL) [10]. This, in turn, encompasses nurses’ satisfaction with their work environment [11,12], work design [13], and the balance between professional and personal responsibilities [9], particularly during crises like the COVID-19 pandemic [14,15].

Quality of work life is a multidimensional concept that reflects nurses’ physical, mental, and emotional well-being in their work environment [16]. A positive QoWL has been linked to better job performance [17], increased job satisfaction [18], and reduced burnout [19], especially in high-stress healthcare settings like those experienced during the COVID-19 pandemic [20].

Preparedness is another essential factor in healthcare, defined as the readiness to respond effectively to crises or emergencies [21]. Preparedness not only enhances nurses’ confidence and competence in managing patient care, but also contributes significantly to their QoWL by reducing workplace stress and promoting job satisfaction [21,22]. Studies indicate that nurses with higher preparedness levels are better equipped to handle the demands of pandemic care, leading to improved resilience and work–life balance [22,23].

For nurses working directly with COVID-19 patients, challenges such as a lack of personal protective equipment (PPE), long working hours, and exposure to suffering have led to compassion fatigue, burnout, and post-traumatic stress. These challenges adversely impact both QoWL and professional performance [24,25,26].

This study explores how EI moderates the relationship between nurses’ preparedness to care for COVID-19 patients and their QoWL. Specifically, it examines whether emotional competencies such as self-awareness, empathy, and relationship management can buffer workplace stress and enhance well-being. Given the intense pressures of the COVID-19 pandemic, investigating these dynamics provides critical insights into how strengthening EI can support nurses’ readiness and foster a more sustainable quality of work life.

### 1.1. Aim

This study aims to explore how emotional intelligence (EI) moderates the relationship between nurses’ preparedness to care for COVID-19 patients and their quality of work life.

### 1.2. Research Questions

What is the level of emotional intelligence among nurses working during the COVID-19 pandemic?What is the reported level of nurses’ preparedness to care for COVID-19 patients?How do nurses perceive the quality of their work life during the pandemic?Is there a relationship between emotional intelligence, readiness to care for COVID-19 patients, and quality of work life among nurses?

## 2. Theoretical Framework

The current study is grounded in the Job Demand–Control (JDC) model, originally proposed by Karasek [27]. This model posits that the interplay between job demands—such as workload, emotional stressors, and physical challenges—and job control, which refers to an employee’s autonomy and decision-making authority, is a key predictor of well-being and performance. The model highlights the importance of creating environments where high job demands are balanced by adequate control, enabling employees to manage stress effectively and sustain productivity.

Social support, particularly from peers and supervisors, is an integral component of this framework, acting as a buffer against the negative effects of high job demands. In nursing contexts, social support enhances teamwork and emotional resilience, both of which are critical during crises like the COVID-19 pandemic. This aligns with the study’s focus on preparedness and EI, as these attributes contribute to a nurse’s ability to manage high job demands while maintaining quality of work life.

In this study, emotional intelligence (EI) is conceptualized as a moderating factor that enhances nurses’ ability to control job demands through its components of self-awareness, self-regulation, empathy, and social skills. By fostering emotional resilience and improving interpersonal dynamics, EI is expected to mitigate the adverse effects of high job demands and improve nurses’ quality of work life [28], particularly in high-stress environments like those encountered during the pandemic (refer to Figure 1).

## 3. Conceptual Framework

This study hypothesizes that emotional intelligence moderates the relationship between nurses’ preparedness to care for COVID-19 patients and their quality of work life (QoWL) (Figure 1). Preparedness, operationalized as the confidence and readiness to deliver patient care during the pandemic, interacts with emotional intelligence to influence nurses’ experiences of their work environment.

The conceptual framework draws from the JDC model by incorporating both job demands (e.g., managing COVID-19 patients) and job resources (e.g., emotional competencies, training, and institutional support). EI is positioned as a resource that not only strengthens nurses’ preparedness, but also buffers the emotional and psychological challenges associated with providing patient care during crises.
The framework integrates three key constructs:
***Emotional Intelligence (EI)*** *(moderating variable): Refers to the nurses’ ability to manage and regulate emotions, hypothesized to influence the relationship between preparedness and quality of work life.***Preparedness to Care for COVID-19 Patients** *(independent variable): Represents the readiness of nurses to effectively manage COVID-19-related care demands.***Quality of Work Life (QoWL)** *(dependent variable): Captures nurses’ overall job satisfaction, emotional well-being, and work–life balance.*

These constructs interact dynamically, with EI enhancing preparedness and QoWL by enabling nurses to cope with stress, build collaborative relationships, and maintain emotional balance in challenging work environments. This interplay underscores the importance of targeting EI in interventions aimed at improving both preparedness and QoWL in nursing practice. The hypothesis for this framework is as follows:

**Hypothesis** **1.**Emotional intelligence (EI) will positively impact nurses’ quality of work life (QoWL) both directly and indirectly by moderating the relationship between nurses’ preparedness to care for COVID-19 patients and their QoWL (refer to Figure 1).

## 4. Literature Review

### 4.1. Emotional Intelligence and QoWL

In the healthcare sector, emotional intelligence (EI) is broadly defined as the ability to recognize, understand, manage, and influence one’s own emotions, as well as the emotions of others [28]. In healthcare settings, where stress is often high, and relationships with patients and colleagues are crucial, EI has emerged as a critical factor influencing job satisfaction [12,29], interpersonal relationships [29], patient outcomes [30], and nurses’ overall well-being [12,31].

Studies in diverse contexts highlight the value of EI in improving nurses’ QoWL. For instance, Tajigharajeh et al. [10] found that higher EI levels are associated with better interpersonal sensitivity and work–life balance, leading to reduced emotional exhaustion. Similarly, Sun et al. [15] demonstrated that Saudi nurses with higher EI experienced lower burnout levels and better QoWL, particularly during the pandemic. These findings underscore the role of EI as a critical resource for enhancing nurses’ well-being in high-pressure environments.

In the regional context, Othman et al. [13] investigated the relationship between EI and organizational commitment among nurses in Qatar. Their findings suggest that EI plays a crucial role in fostering resilience and commitment among nurses facing systemic challenges in the Gulf region. Nationally, Saudi Arabian studies by Aljarboa et al. [32] and Shahin et al. [33] corroborate these findings, demonstrating that nurses with higher EI levels reported improved stress management, lower burnout, and higher QoWL during the COVID-19 pandemic.

### 4.2. Preparedness and QoWL

Preparedness is an essential concept in healthcare, particularly in response to pandemics and other health crises. Defined as the readiness to perform tasks under crisis conditions [21], preparedness directly influences nurses’ confidence, mental resilience, and QoWL [22,23]. According to the Theory of Planned Behavior (TPB), preparedness is shaped by attitudes, subjective norms, and perceived behavioral control [34].

International studies, such as those conducted by Razu et al. [24], and Hill et al. [35] highlighted global challenges during crises like the COVID-19 pandemic, including shortages of protective equipment and inadequate training, which significantly impacted nurses’ sense of preparedness during the pandemic. However, supportive environments that addressed these barriers improved both preparedness and job satisfaction.

National studies, including those by Altwaijri et al. [36], Sheerah et al. [26], Alharbi et al. [37], and Al Mutair et al. [16] underline similar challenges and emphasize the need for enhanced training and psychological support to ensure effective responses to future crises. Moreover, they linked higher preparedness levels among nurses to better QoWL and lower stress levels during the pandemic. These findings suggest that fostering preparedness through training and resource availability can mitigate the negative impacts of crises on nurses’ well-being.

### 4.3. Articulating EI, QoWL, and Preparedness

The interplay between emotional intelligence (EI), preparedness, and quality of work life (QoWL) has garnered increasing attention in the nursing literature. EI enables nurses to effectively regulate their emotions, which contributes to a greater sense of preparedness when managing complex and stressful healthcare demands. Preparedness, in turn, has been linked to nurses’ confidence in managing patient care, ultimately influencing their QoWL. Studies highlight that nurses with high EI not only demonstrate superior readiness during crises, but also report enhanced QoWL through improved emotional resilience and reduced stress levels [14,22]. These findings suggest that EI acts as a bridge between preparedness and QoWL, creating a synergistic relationship that strengthens nurses’ professional and personal outcomes during challenging situations.

Streamlining the relationship between emotional intelligence (EI), preparedness, and quality of work life (QWL) among nurses, particularly in crises such as the COVID-19 pandemic, highlights their interconnected nature. Nurses equipped with emotional intelligence are not only better prepared to handle crises, but also more likely to report a positive QoWL. Both regional and national studies highlight the buffering effect of EI in mitigating stress and enhancing resilience, thereby improving nurses’ preparedness and work–life balance, even in the face of insufficient resources. This review underscores the need for further investigation into how EI can be leveraged to support nursing staff, aligning with the aim of this study, which seeks to explore the moderating role of emotional intelligence in the relationship between preparedness to care for COVID-19 patients and the quality of work life among nurses.

## 5. Materials and Methods

### 5.1. Study Design

The study design is cross-sectional, descriptive, and correlational, conducted across multiple healthcare sites. A convenience sample was used to assess the moderating effect of emotional intelligence (EI) on the relationship between nurses’ preparedness to care for COVID-19 patients (independent variable) and their quality of work life (QWL) (dependent variable).

### 5.2. Participants

A convenience sample of registered nurses working in inpatient wards, emergency departments, and critical care units across multiple healthcare settings who met the inclusion criteria were invited to participate. Inclusion criteria included full-time bedside nurses with at least one year of experience in the selected settings, proficiency in English, and willingness to participate in the study. The sample size was estimated at 267 nurses using the G*power program.

### 5.3. Study Instrument

Data were collected using an electronic survey that included three parts:Demographic and work-related variables: variables included age, gender, marital status, educational level, nationality, work location, working unit, job title, and working hours per week.Emotional Intelligence Scale (EI (PcSc) Scale): The EI of participants was measured using the EI (PcSc) Scale by Mehta and Singh [38] based on Goleman’s Emotional Intelligence Competency Model [28]. This 69-item self-report scale measures EI with two subscales: personal competence and social competence, and scores range from 1 (extremely low competence) to 5 (extremely high competence), with higher scores indicating greater emotional intelligence. The scale has demonstrated strong reliability (Cronbach’s alpha = 0.91).Quality of Nursing Work Life (QNWL) Survey: A modified version of Brooks’ Quality of Nursing Work Life Survey [39] assessed nurses’ perceptions of their work life quality. This 42-item survey covers four dimensions: work life, work context, work design, and work world, rated on a 5-point Likert scale (1 = completely disagree, 5 = completely agree). For this study, Cronbach’s alpha was 0.95, indicating excellent reliability.Preparedness to Care for COVID-19 Patients: preparedness was assessed with several items that measured nurses’ perceived readiness and confidence in caring for COVID-19 patients, including attendance at COVID-19 workshops, direct patient interaction, and perceived adequacy of pandemic support. Responses to these items were combined to create a preparedness score, where higher values indicate greater preparedness.

A pilot study with 12 participants was conducted, and the survey was reviewed by eight expert reviewers to ensure its clarity and validity.

### 5.4. Ethical Considerations

Ethical approval was obtained from the Institutional Review Board (IRB) Health Sciences Colleges Research on Human Subjects, King Saud University and the Standing Committee for Research Ethics on Living Creatures in Ministry of Health,. Informed consent was collected electronically before survey participation, ensuring that participants were fully aware of the study’s purpose and procedures, and their rights. The anonymity and confidentiality of participants were strictly maintained throughout data collection and analysis. All data were stored securely and encrypted, and were accessible only by the research team.

For electronic surveys, additional ethical measures were taken to ensure participant privacy. Each participant was assigned a unique identifier, and responses were stored in a secure cloud-based platform compliant with regional and institutional data protection regulations. The study followed ethical standards consistent with the Declaration of Helsinki.

### 5.5. Procedure for Data Collection

Data collection was conducted through an electronic survey (e-survey) distributed to participants via a secure online platform. Before the administration of the e-survey, the necessary permissions were obtained:Adoption of Tools: Approval was sought from the original authors of the Emotional Intelligence (EI) Scale and Brooks’ Quality of Nursing Work Life Survey to use and adapt their instruments for this study. Correspondence was initiated, and written consent was obtained, ensuring adherence to copyright regulations.Ethical Approval: Ethical clearance was obtained from the Research Ethics Committees in the Ministry of Health, ensuring compliance with national ethical guidelines for research involving human subjects. The research proposal was submitted for review, and feedback was incorporated before final approval was granted.Administrative Permissions: Official approval was secured from the administration of each participating hospital. This involved presenting the study’s objectives, methodology, and potential benefits to nursing staff and hospital management to ensure support and cooperation during the data collection process.

Following these approvals, the e-survey was distributed to registered nurses meeting the inclusion criteria. Participants received an invitation via email, which included a brief overview of the study, an assurance of anonymity, and a link to access the e-survey. Consent was obtained electronically before participants could proceed to complete the survey.

To ensure a high response rate, reminders were sent to participants one week after the initial invitation, encouraging their participation and emphasizing the importance of their input in enhancing nursing practices and policies.

### 5.6. Statistical Analysis

Data were analyzed using IBM SPSS Statistics version 27. Descriptive statistics, including frequencies and percentages, means, and standard deviations, were calculated to describe the sample characteristics and the responses to key survey variables, including emotional intelligence levels, preparedness to care for COVID-19 patients, and quality of work life. Preparedness was operationalized as the nurses’ self-reported readiness and access to necessary resources and training for COVID-19 patient care. Preparedness scores were calculated based on participants’ responses to specific items, assessing their perceived readiness, available training, and support systems. Pearson’s correlation coefficient was used to explore the relationship between emotional intelligence, preparedness, and quality of work life. Multiple regression analysis was employed to assess whether emotional intelligence moderates the relationship between preparedness and QWL. Interaction terms were created for moderation testing. A significance level of *p* < 0.05 was set for all analyses.

## 6. Results

### 6.1. General Characteristics:

The study sample comprised 267 participants, predominantly female (94.4%), with a mean age of 37.47 years (SD = 8.09) and an average of 8.43 years of experience (SD = 6.33). Of the participants, 59.6%were married, 68.5% held a bachelor’s degree in nursing, and 79% were non-Saudi. The bedside nurses comprised 88.4% of the sample, and 82% worked more than 40 h per week. For further demographic details, see Table 1.

### 6.2. Research Question 1: What Is the Level of Emotional Intelligence Among Nurses Working During the COVID-19 Pandemic?

The emotional intelligence (EI) of nurses, assessed through personal and social competence subscales, showed high overall levels. Among participants, 93.6% reported a “good” level of personal competence, and 85.4% reported a “good” level of social competence. Subcomponents like self-awareness and social skills scored particularly high, with over 80% of nurses rating themselves positively in these areas. These findings suggest that nurses in this study demonstrated strong EI, which may support their ability to manage stress effectively in high-pressure environments (See Table 2 for summary; for detailed data, please see Appendix A).

### 6.3. Research Question 2: What Is the Reported Level of Nurses’ Preparedness to Care for COVID-19 Patients?

Preparedness to care for COVID-19 patients had a mean score of 2.98 (SD = 0.32) on a 3-point scale, indicating a high level of readiness among nurses. Approximately 76.4% of nurses reported feeling prepared, while 13.5% reported feeling unprepared, and 10.1% were undecided.

Additionally, 87.3% of nurses attended workshops related to COVID-19 care, and 84.3% reported direct interaction with COVID-19 patients. However, only 72.7% indicated receiving adequate support during the pandemic. However, perceived stress of providing care varied, with 44.2% rating it as very stressful, and 4.5% finding it not stressful (see Table 3).

### 6.4. Research Question 3: How Do Nurses Perceive the Quality of Their Work Life During the Pandemic?

Overall, 65.5% of nurses rated their QoWL as “good”, and the remaining 34.5% perceived it as “fair”. Across specific subscales, “work design” and “work context” received the most favorable ratings, with over 75% indicating positive perceptions. “Home life/work life” and “work world” showed more varied ratings, reflecting both positive and moderate perceptions. These results suggest that, even amid pandemic conditions, a supportive work environment can positively influence nurses’ QoWL (See Table 4 for summary; for detailed data, please see Appendix A).

### 6.5. Research Question 4: Is There a Relationship Between Emotional Intelligence, Readiness to Care for COVID-19 Patients, and Quality of Work Life Among Nurses?

Pearson correlation analysis indicated a significant positive correlation between personal competence (r = 0.33, *p* < 0.001) and social competence (r = 0.34, *p* < 0.001) with the quality of work life. This indicates that nurses with higher levels of EI, both personal and social, experienced a better QoWL. There was no significant correlation with either emotional intelligence or quality of work life (see Table 5). This result suggests that while EI strongly influences QoWL, preparedness alone may not directly impact QoWL.

### 6.6. Hypothesis Testing: Does Emotional Intelligence Moderate the Relationship Between Readiness to Care for COVID-19 Patients and Quality of Work Life Among Nurses?

“Emotional intelligence (EI) will positively impact nurses’ quality of work life (QoWL) both directly and indirectly by moderating the relationship between nurses’ preparedness to care for COVID-19 patients and their QoWL”.

A hierarchical regression analysis was performed to examine whether emotional intelligence moderates the relationship between preparedness and QoWL. The initial model showed that preparedness to care for COVID-19 patients was not a significant predictor of quality of work life (β = 0.034, *p* = 0.57). However, when including interaction terms, the model explained 41% of the variance in quality of work life, indicating that significant moderation effects (personal competence: β = 0.578, *p* < 0.001; social competence: β = 0.665, *p* < 0.001) were found to buffer the relationship, suggesting that higher EI levels mitigate the potential stressors of COVID-19 patient care, as summarized in Table 6 and Figure 2.

## 7. Discussion

The findings of this study underscore the critical role of emotional intelligence (EI) and preparedness in shaping nurses’ quality of work life (QoWL) during the COVID-19 pandemic. The study demonstrates how EI acts as a moderating factor, highlighting its potential to mitigate workplace stress and enhance resilience among nurses. These findings align with the broader literature that underscores the EI’s importance in healthcare, particularly during high-pressure situations such as global health crises [10,11].

Our findings indicate that nurses exhibited high levels of both personal and social competencies, consistent with previous research linking EI to better coping mechanisms and professional satisfaction [40]. For instance, Moradian et al. [14] found that nurses with higher EI displayed better emotional regulation and resilience during stressful patient interactions, reducing their risk of burnout and compassion fatigue. In the context of COVID-19 care, such competencies are particularly crucial, as they enable nurses to manage the psychological burden of frontline care. Similarly, Karimi et al. [8] noted that emotional awareness and self-regulation among healthcare professionals enhance mental well-being and performance under stress. However, discrepancies with findings from Wuhan [41], where varying levels of EI were observed alongside significant anxiety and stress [15], may reflect differences in institutional support systems. These findings emphasize the role of supportive environments in enhancing EI and resilience, particularly in regions with structured mental health resources and managerial backing [15,32,41].

Regarding preparedness to care for COVID-19 patients, most nurses reported feeling prepared, reflected in their high self-reported readiness scores. This aligns with previous studies that highlight the role of organizational preparedness and training in boosting healthcare workers’ confidence during crises [42]. Goniewicz et al. [22] demonstrated that organizational preparedness positively influences nurses’ perceived readiness, further enhancing their ability to manage stress and avoid burnout. In contrast, other studies, such as those by Razu et al. [24], Gordon et al. [43], and Marcomini et al., [44] identified significant stress and trauma among healthcare workers during early pandemic waves, particularly in settings with insufficient resources [43,44]. These variations may be attributed to the presence of comprehensive training programs and psychological support systems in our sample, which bolstered nurses’ preparedness to face the challenges of caring for COVID-19 patients [23]. For instance, simulation training and workshops designed for pandemic care may have provided nurses with the tools needed to navigate the complexities of patient care confidently [45].

The majority of participants reported a positive QoWL, corroborating findings from studies that highlight the benefit of supportive work environments on employee morale [20]. In the Saudi Arabian context, Baghdadi et al. demonstrated that nursing students with high EI exhibited better emotional regulation and caring behaviors, suggesting that EI enhances resilience and supports work life quality [46]. Nevertheless, findings from other healthcare settings highlight the pandemic’s exacerbation of burnout and compassion fatigue among nurses, highlighting the complexity of the work life experience during this period [33,45,47]. For example, institutions offering mental health check-ins, flexible scheduling, and peer support were more likely to maintain positive QoWL outcomes compared to those lacking such measures [48].

The interaction between EI, preparedness, and QoWL revealed significant correlations in our study, reinforcing the importance of emotional and social competencies for improved work outcomes. This is supported by Pérez-Fuentes et al. [49], who observed that nurses with higher EI demonstrated greater engagement and job satisfaction. High social competence fosters collaboration, which can enhance the quality of patient care and reduce workplace stress [14]. Similarly, studies by Othman et al. [13] in Qatar emphasized that nurses with elevated EI exhibited higher resilience and organizational commitment, particularly during pandemic-related pressures [50,51]. However, the lack of a significant correlation between preparedness to care for COVID-19 patients and either of the two forms of EI suggests a nuanced relationship. This finding may reflect the emotional burden of pandemic care, where perceived readiness may not fully mitigate stress or feelings of inadequacy during crises [52].

The moderating role of EI in the relationship between preparedness and QoWL is a key finding, highlighting its protective effect against stress during crisis care. Aljarboa et al. [32] and Turjuman and Alilyyani [9] similarly emphasized that nurses with higher EI demonstrated better work engagement and adaptability, even when facing limited institutional support. Targeted interventions to enhance EI, such as emotional regulation training [29], could equip nurses with the tools needed to manage challenging patient interactions and reduce stress [53]. This underscores the need for organizational strategies that prioritize EI development as a means to foster a resilient nursing workforce [54].

### 7.1. Implications of the Study

The findings of this study underscore several important implications for nursing practice and workforce well-being, particularly in high-stress settings such as those experienced during the COVID-19 pandemic:

Emotional Intelligence as a Core Competency: given the significant role of Emotional Intelligence (EI) in enhancing nurses’ quality of work life (QoWL), healthcare institutions should consider integrating EI development programs into nursing training and professional development. The training focused on self-awareness, emotional regulation, and interpersonal skills, which could help nurses build resilience, manage stress, and maintain well-being, which are critical for sustaining performance under pressure.

Strengthening Preparedness through EI: the study highlights the fact that EI serves as a moderating factor that enhances the effects of preparedness on QoWL. This suggests that EI-focused interventions can bolster nurses’ confidence and effectiveness in handling crises. Preparedness programs that integrate EI skills may enable nurses to better manage the emotional challenges of pandemic care, improve job satisfaction, and reduce burnout risk.

Enhancing Support Systems: although this study focused on EI and preparedness, the findings imply that broader support systems are essential for maximizing EI’s positive effects. Institutional support, such as providing access to mental health resources, peer support, and sufficient PPE, can reinforce nurses’ sense of preparedness and well-being.

Establishing a supportive work environment can, therefore, amplify the benefits of high EI, improving retention rates and job satisfaction in the nursing workforce.

Policy Development and Organizational Change: our findings advocate for organizational policies that prioritize EI as part of the nursing workforce’s professional competency framework. By embedding EI into institutional culture and offering regular training opportunities, healthcare leaders can foster a work environment where nurses feel emotionally supported and prepared to face the challenges of patient care, leading to enhanced QoWL and reduced turnover intentions.

Future Research and Practice: this study emphasizes the importance of exploring how EI interacts with other organizational factors, such as stress and institutional support, which were outside the current scope. Future studies examining these interactions could provide a more comprehensive understanding of the influences on QoWL, guiding strategies for cultivating a resilient and well-prepared nursing workforce.

These implications highlight the potential of EI development as an effective strategy for supporting nurses’ mental health, job satisfaction, and resilience, ultimately contributing to a sustainable nursing workforce in challenging healthcare settings.

### 7.2. Limitations

Despite the valuable insights gained from this study, several limitations must be acknowledged. First, the cross-sectional design limits the ability to infer causality between emotional intelligence, preparedness to care, and quality of work life. This limitation impacts our results by preventing the identification of potential changes in EI or preparedness over time, particularly as nurses gain experience or encounter prolonged crises. Longitudinal studies could provide a deeper understanding of how these factors evolve, particularly during prolonged crises. Additionally, the integrative qualitative component would allow for a deeper exploration of nurses’ lived experiences, further clarifying how EI and preparedness impact work life quality in crises contexts.

Second, the study relied on self-reported measures, which may be subject to bias, including social desirability bias, which could affect the accuracy of responses. Self-reporting may lead to participants presenting themselves in a more favorable light, potentially overstating levels of EI or preparedness. Future research could address this by employing mixed methods or incorporating structured interviews to validate and enrich self-reported data. Specific geographic regions, with distinct healthcare challenges, which may limit the generalizability of the findings to other regions or healthcare settings with different demands and resources, may mean that the positive effects observed could be unique to the facilities surveyed, rather than universally applicable. This regional focus was chosen due to logistical constraints and the region’s significant impact from COVID-19; however, expanding the geographic scope in future studies would enhance generalizability and provide insights into how regional and institutional differences influence the dynamics of EI, preparedness, and work life quality.

Finally, while the study focused on emotional intelligence, other potentially influential factors, such as organizational culture, support systems, and external stressors, were not examined in depth. These factors might interact with EI and preparedness, potentially influencing the quality of work life. Future studies should explore these additional factors, as they likely play critical roles in shaping the work life quality of healthcare professionals, especially during crises.

## 8. Conclusions

This study underscores the importance of emotional intelligence as a moderating factor in the relationship between nurses’ preparedness to care for COVID-19 patients and their quality of work life. The findings suggest that fostering EI among nursing professionals can significantly enhance their resilience, coping strategies, and overall job satisfaction, particularly in high-stress environments like those experienced during the pandemic.

To maximize the benefits of EI for nursing staff, healthcare institutions should consider implementing EI training programs as part of professional development and ongoing education. Specific recommendations for these programs include the following:

Incorporating EI into Nursing Curricula and Professional Development: institutions should integrate EI competencies, such as self-awareness, self-regulation, empathy, and interpersonal skills, into both nursing education and continuing professional development programs. This could be achieved through workshops, simulation training, and reflective practice sessions that provide nurses with tools to manage emotions effectively.

Offering Regular EI-Focused Workshops and Support Groups: regular workshops that focus on building emotional and social competencies can provide nurses with strategies to better manage stress and enhance their readiness to handle crisis situations. Support groups or peer mentoring programs could further reinforce these skills by creating safe spaces for nurses to share experiences and discuss challenges.

Evaluating and Sustaining EI Training Programs: to ensure effectiveness, healthcare institutions should establish metrics to evaluate the impact of EI training on nurses’ QoWL, resilience, and patient care outcomes. Regular assessments and feedback from nursing staff can help refine these programs, making EI a sustainable and adaptive component of nursing practice.

While many nurses reported a positive quality of work life, the interplay of emotional competencies and preparedness to care reveals a complex relationship that warrants further exploration. Future research should aim to address the limitations identified in this study and explore additional factors that may influence the well-being of nurses in challenging healthcare settings. By prioritizing emotional intelligence training and support, healthcare organizations can better equip nurses to face the multifaceted challenges of patient care during crises, ultimately improving both nurse and patient outcomes.

## Figures and Tables

**Figure 1 behavsci-14-01166-f001:**
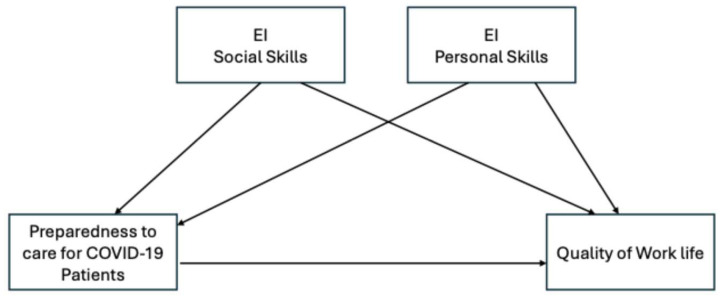
Conceptual Framework.

**Figure 2 behavsci-14-01166-f002:**
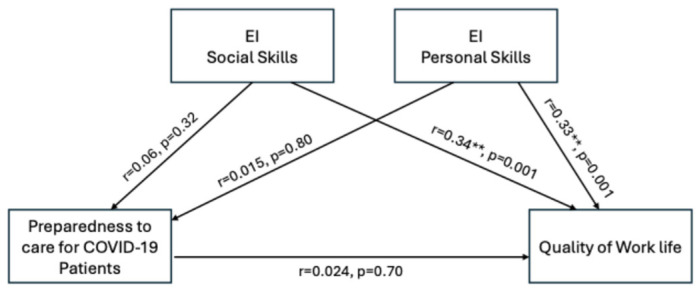
The moderating role of emotional intelligence between nurses’ preparedness to care for COVID-19 patients and their quality of work life. ** *p* < 0.01.

**Table 1 behavsci-14-01166-t001:** Demographic and work-related characteristics of the study participants (N = 267).

Variable		Mean	SD
Age (Yrs)		37.47	8.09
Years of Experience (Yrs)		8. 43	6.33
**Variable**		**Frequency** **(N = 267)**	**Percentage** **(%)**
**Gender**	Men	14	5.6
	Women	252	94.4
**Marital Status**	Single	95	35.6
	Married	159	59.6
	Divorced	8	3
	Widowed	5	1.9
**Educational Level**	Diploma	65	24.3
	Bachelor	183	68.5
	Postgraduate ^1^	19	7.1
**Nationality**	Saudis	56	21
	Non-Saudis	211	79
**Working Unit**	General wards	115	43.1
	Critical Care Units	63	23.6
	Specialized Units	89	33.3
**Job Title**	Bedside nurse	236	88.4
	Nurse managers	31	11.6
**Working hours per week**	More than 40 h	219	82
	From 30 to 40 h	39	14.6
	Fewer than 30 h	9	3.4

^1^ Include Master’s and PhD holders.

**Table 2 behavsci-14-01166-t002:** Level of emotional intelligence among nurses working during the COVID-19 pandemic (N = 267).

Emotional Intelligence Subscales	Frequency (%)			X (SD)
	Good	Fair	Poor	
**Personal competence**	250 (93.6)	17 (6.4)	0 (0)	2.75 (0.41)
Self-Awareness	246 (91.4)	21 (7.8)	0 (0)	2.91 (0.27)
Self-Motivation	222 (82.5)	44 (16.4)	1 (0.4)	2.83 (0.39)
Emotional Regulation	196 (72.9)	64 (25.7)	2 (0.7)	2.73 (0.46)
**Social competence**	228 (85.4)	39 (14.6)	0 (0)	2.83 (0.38)
Social-Awareness	221 (82.2)	46 (17.1)	0 (0)	2.83 (0.38)
Social Skills	336 (87.7)	30 (11.2)	1 (0.4)	2.88 (0.37)
Emotional Receptive	201 (74.7)	64 (23.8)	2 (0.7)	2.74 (0.45)
**Overall Emotional Intelligence**	**223 (83.5)**	**41(15.3)**	**3(1.2)**	**2.83 (0.37)**

**Table 3 behavsci-14-01166-t003:** Preparedness to care for COVID-19 Patients (N = 267).

Variable		Frequency (N = 267)	Percentage (%)
**Attending COVID-19-related workshop(s)**	Yes	233	87.3
	No	34	12.7
**Direct interaction with COVID-19 patients**	Yes	225	84.3
	No	42	15.7
**Received proper support during the pandemic**	Yes	194	72.7
	No	73	27.3
**PPE is a necessity in caring for COVID-19 patients**	Necessary	267	100
	Not necessary	0	0
**Reported level of stress in caring for COVID-19 patients**	Very stressful	118	44.2
	Moderately stressful	92	34.5
	Slightly stressful	45	16.9
	Not stressful	12	4.5
**Overall preparedness**	Ready	239	88.8
	Not decided	27	10.0
	Not ready	1	0.4
	X (SD)	2.98 (0.32)	

**Table 4 behavsci-14-01166-t004:** Nurses’ perception of the quality of their work life during the pandemic (N = 267).

Quality of Work Life Subscales	Frequency (%)			X (SD)
	Good	Fair	Poor	
Home life/work life	145 (53.9)	121 (45.3)	1 (0.4)	2.54 (0.51)
Work design	222 (82.5)	45 (16.9)	0 (0)	2.83 (0.38)
Work context	208 (77.3)	58 (21.6)	1 (0.4)	2.78 (0.43)
Work world	127 (47.2)	131 (48.7)	9 (3.4)	2.44 (0.56)
**Overall QWL**	175 (65.5)	92 (34.5)	0 (0)	2.91 (0.27)

**Table 5 behavsci-14-01166-t005:** Correlation between EI, preparedness, and quality of work life among nurses (N = 267).

Variables	Personal Competence	Social Competence	Preparedness	Quality of Work
**Personal competence**	-			
**Social competence**	r = 0.52 ** *p* = 0.000	-		
**Preparedness**	r = 0.015 *p* = 0.80	r = 0.060 *p* = 0.32	-	
**Quality of work life**	r = 0.33 ** *p* = 0.000	r = 0.34 ** *p* = 0.000	r = 0.024 *p* = 0.70	-

** *p* < 0.001.

**Table 6 behavsci-14-01166-t006:** Regression test of moderation effect of EI on the relationship between preparedness to care for patients with COVID-19 and quality of work (N = 267).

Regression	B	SEB	β	*p*
**Preparedness**	−0.487	0.081	−1.17	0.001
**Personal competence x preparedness**	0.080	0.021	0.578	0.001
**Social competence x preparedness**	0.094	0.032	0.665	0.001
r = 0.41 **		*p* = 0.001		

** *p* < 0.001.

## Data Availability

The corresponding author is willing to provide the data that support this study’s findings, upon receiving a reasonable request.

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
