# Peer review of "The Moderating Role of Emotional Intelligence on the Relationship Between Nurses’ Preparedness to Care for COVID-19 Patients and Their Quality of Work Life"

_behavsci, 2024, doi:10.3390/bs14121166_

Round 1

Reviewer 1 Report

Comments and Suggestions for Authors

The manuscript under evaluation offers a perspective of great value to the journal's audience. It focuses on the role of Emotional Intelligence (EI) in the quality of professional life within the field of nursing in the pandemic. The results presented are interesting, significant and contribute to previous literature in the field of EI and psychological well-being within the field of health. In general, the article presents several strong points, it is a growing topic and of great interest today. Furthermore, the analyzes are strong, in general, and allow relevant conclusions to be drawn. The article is written in a clear and organized manner, allowing the reader to correctly understand it in its entirety. Despite these strengths, there are some weaknesses that must be addressed. Below is an assessment of each section detailing the changes that must be made before publication:

The abstract is correct but should be somewhat more precise and clear, it does not clearly detail the key results.

The introduction provides a correct overview of the context of the pandemic and details the challenges, which allows us to glimpse the relevance of the study. Furthermore, the objectives are clear and formulated in a feasible way, as is the hypothesis. Despite this, although the context is presented, a more in-depth review of previous studies that relate EI to quality of work life in the field of nursing is missing, which could strengthen the justification of the study. A more comprehensive review of the existing literature should be included to better support the need and relevance of the study.

The methodology is described in detail, meets ethical considerations and presents a valid study. However, the sample size, although significant, is limited to a specific region, which may limit the generalizability of the results. The choice of sample should be justified and a more geographically diversified approach should be considered in future research.

The results are presented in a complete, coherent and organized manner. The statistical analyzes carried out are correctly detailed. Despite this, although a descriptive analysis has been carried out, an additional, more in-depth analysis could have been done that explored factors that may be impacting EI and quality of professional life, such as stress and support from the institution. It is requested to expand the discussion of results to address the implications of the findings in nursing practice.

The discussion adequately contextualizes the findings within the existing literature. The importance of EI in nursing practice is emphasized, which may have significant practical implications. However, the discussion is somewhat general and could be improved with concrete examples or case studies that illustrate the relationship between EI and nurses' well-being. It is necessary, therefore, to integrate specific examples or include references to previous studies that demonstrate the positive impact of EI in the work environment of nurses.

Important limitations that affect the validity and generalizability of the study, which is essential in scientific research, are addressed. Although these limitations are acknowledged, the discussion could be more specific about how each limitation impacts the results. It is requested to provide examples of how these limitations could be mitigated in future (prospective) research.

The conclusions are consistent with the findings and highlight the importance of EI in the context of nursing during the pandemic. Despite this, they could be more specific about practical recommendations for EI training in the nursing context. It would be appropriate to include some specific recommendations for the implementation of EI training programs within health institutions.

Reviewer 2 Report

Comments and Suggestions for Authors

The study is interesting, relevant and could present important contribution to the research of EI in the healthcare context. However, substantial improvement needs to be made.

Introduction

In general, the introduction should be clearer, better structured and more concise. It is not common that descriptions of the core concepts of investigation follow research questions. Research question and hypotheses should be placed in the end of the introduction section. Also, there is much repetitions in the description of relations of emotional intelligence with different aspects of nursing, and, in the same time, construct of emotional intelligence is poorly explained. The same is true for the discussion related to relations between preparedness and QWL. In my opinion there is too much discussion on the healthcare system throughout the introduction, that doesn't contribute to the understanding of the research questions. Furthermore, the theories that are relevant for the research should not be mentioned in passing. Theoretical background of the conceptual framework should be better explained. Refferences are missing in many places of the introduction.

Method section

It is not clear why authors state that this is multi-site and descriptive study. Variables are poorly described. It is not clear how the variable of preparedness was operationalized and calculated. 

Results

The results are not well connected to the research questions and hypotheses. The same variables are not present in all parts of the manuscript. Descriptive statistics of the variables is not provided. 

Discussion should be adapted to changes in other part of the manuscript.

I suggest English proofreading.

Comments on the Quality of English Language

I suggest English proofreading.

Reviewer 3 Report

Comments and Suggestions for Authors

Dear authors,

It was with great pleasure that I reviewed your manuscript, on emotional intelligence in nurses in times of pandemic.It is a very pertinent topic, which, despite being relevant in times of pandemic, remains current and should remain in the interests of nursing professionals.

The title is concise, clear and adequately describes the content of the article.

Summary: indicates the methodology, research question, theory, method, main results and implications. 

Conceptual development and hypotheses. Hypotheses predict some type of relationship between variables. The authors have included a description of the instruments along with an explanation of the variables, allowing readers to understand what is being measured and how. As expected, we have an independent variable and a dependent variable, in that order.

Methods: Validated regarding data collection procedures and instruments, sample, variables and data analysis procedures.

 Results. The authors included descriptive elements to observe the distribution of data and identify the association between variables, as well as the usual measures of collinearity. They contain tables with statistical results and indicate the verification of the hypothesis.

Discussions and conclusions. They mention the objective of the article and how it was achieved. They talk about the implications and contributions and an analysis of the main results in relation to the exposed theory.

They present limitations of the study and suggestions for future research. All these elements flow coherently and without deviating into secondary issues.

However, they need to review the key words, which are not all validated in Decs/mech, and review the citations and bibliographic references, which are not uniform.

                  232 / 5 000  

Resultados da tradução

Resultado de tradução

Literature review: the authors not only present a collection of previous works, but make the connection between previous works and the present. Demonstrate that they know and understand the fundamentals that underlie EI.

Round 2

Reviewer 1 Report

Comments and Suggestions for Authors

The authors have responded correctly to the instructions offered

Author Response

Thank you for your positive feedback and acknowledgment that the revisions have correctly addressed the instructions offered. We appreciate your thorough review and are grateful for your guidance throughout this process.

Reviewer 2 Report

Comments and Suggestions for Authors

Introduction, as well as other parts of the manuscript, were not improved enough. Authors have not considered all comments from previous review, and have not done enough research of the different research questions, so introduction is not shaped in the satisfactory way and references are not appropriate. Authors haven't accepted comments on the structure of the introduction. In the different parts of the manuscript not all variables were mentioned and descriptive statistics of the variables were not reported. All mentioned criteria are very important for the quality of the manuscript, so I think that this manuscript is not ready for the publication.

Comments on the Quality of English Language

I suggest English proofreading.
